# Dynamic Error Correction of Filament Thermocouples with Different Structures of Junction based on Inverse Filtering Method

**DOI:** 10.3390/mi11010044

**Published:** 2019-12-30

**Authors:** Chenyang Zhao, Zhijie Zhang

**Affiliations:** 1School of Instrument and Electronics, North University of China, Taiyuan 030051, China; lyfcnzcy@163.com; 2Key Laboratory of Instrumentation Science & Dynamic Measurement (North University of China), Ministry of Education, Taiyuan 030051, China

**Keywords:** filament thermocouple, 1D inverse heat conduction, regularization, filtering kernel, two-layer domain

## Abstract

Since filament thermocouple is limited by its junction structure and dynamic characteristics, the actual heat conduction process cannot be reproduced during the transient thermal shock. In order to solve this problem, we established a thermocouple dynamic calibration system with laser pulse as excitation source to transform the problem of the restoring excitation source acting on the surface temperature of thermocouple junction into the problem of solving the one-dimensional (1D) inverse heat conduction process, proposed a two-layer domain filtering kernel regularization method for double conductors of thermocouple, analyzed the factors causing unstable two-layer domain solution, and solved the regular solution of two-layer domain by the filtering kernel regularization strategy. By laser narrow pulse calibration experiment, we obtained experimental samples of filament thermocouples with two kinds of junction structures, butt-welded and ball-welded; established error estimation criterion; and obtained the optimal filtering kernel parameters by the proposed regularization strategy, respectively. The regular solutions solved for different thermocouples were very close to the exact solution under the optimal strategy, indicating that the proposed regularization method can effectively approach the actual surface temperature of the thermocouple junction.

## 1. Introduction

Filament thermocouple has as characteristics simple structure, fast dynamic response, wide range of temperature measurement, ability to measure the ‘single point’ temperature, and its electrical signal is easy to access to the test system, so it has been widely applied in industrial production and scientific research, such as wind tunnel heat flow experiment and explosive thermal shock experiment. Therefore, it is very important to accurately obtain the temperature of the heat sources during the actual tests [1,2,3,4]. Whether there is a wind tunnel environment or an explosion field one, the heat source is usually a directional narrow pulse signal. Its rising time can reach the level of microseconds, and pulse width is in milliseconds. The response of thermocouple to it is impulse response and limited by the dynamic characteristics of the thermocouple itself, such as the thermal inertia of temperature sensing element. Its dynamic response process cannot accurately recover the thermal shock process, which generates great dynamic error [5]. Therefore, it is necessary to analyze whether thermocouple’s dynamic characteristics are competent for the transient temperature test.

The temperature measurement accuracy of the thermocouple is directly affected by the junction welding quality [6]. For filament thermocouple to reduce the heat conduction error and dynamic response error, the structures of junction welded are different. The frequently-used structures are butt-welded and ball-welded as shown in Figure 1a,b.

We can establish a dynamic calibration system with laser pulse as excitation source [7,8,9], excite the thermocouple with a narrow pulse signal, and establish an inverse mathematical model of thermocouple by means of the inter-conversion for the input/output relationship of the thermocouple. Many scholars describe this model as a transfer function, and nonlinear system identification is often used to build a mathematical model to correct the dynamic error with the output of this model and recover the temperature signal at the thermocouple junction. The problem lies in neglecting the physical process of thermocouple heat conduction; thus, the models built are unpractical for engineering applications with large temperature range [10,11,12]. Therefore, from the perspective of inverse filtering correction, the problem of inverse mathematical model identification for the thermocouple can be transformed into the problem of the solution of the process of inverse heat conduction [13].

In inverse heat conduction, our purpose is to correct the impulse response error of thermocouple to approach the actual surface temperature of thermocouple junction. The thermocouple is composed of two metal conductors made of different materials; thus, the two conductors produce two thermal properties during heat conduction. According to the working principle of thermocouple, the thermoelectric potential of the thermocouple is determined by the difference in the temperature distribution function between the two ends of the thermocouple. Therefore, we define the temperature distribution functions at both ends of the thermocouple. The measured output thermoelectric potential difference of thermocouple during thermal shock can be used to solve the surface temperature of thermocouple junction. Therefore, this process can be called the non-characteristic Cauchy problem during heat conduction [14,15]. Any tiny measurement error in the non-characteristic Cauchy problem of heat equation may cause a large error of solution. The continuous dependence of Cauchy numerical solutions were obtained under additional conditions. This is known as conditional stability [16]. In addition, the stable solution of ill-posed problem can be theoretically reconstructed under priori conditions. Due to the severity of this problem, no satisfactory stable approach was not obtained for the Cauchy problem of most classical heat conduction equations, then some regularization strategies are needed [17]. For the 1D inverse heat conduction in monolayer domain, theoretical study and computational implementation have been well studied [18,19], but for thermocouple structure, the problem is transformed into complex inverse heat conduction in two-layer domain. We can decompose multilayer inverse heat conduction into multiple monolayer problems and solve each monolayer problem layer by layer by using the relationship between the layers.

The filament thermocouple was used as a two-layer volume; the junction was defined as the interface between two layers; the heat flow direction is perpendicular to the thermocouple junction; the two metal conductors were defined as of equal length, and one of them is as the free end that its boundary temperature is 0 °C. See Figure 2.

Here, l is the length of the two conductors of the thermocouple (the free end is the first layer; then, the initial value of the working end is l); $R_{1}$ and $R_{2}$ are the regions of the free end and the working end, respectively.

Let us say that $k_{1}$, $k_{2}>0$ are the thermal conductivities of the thermocouple’s two conductors, and $\alpha_{1},\alpha_{2}>0$ are the thermal diffusion coefficients of the first layer and the second layer, respectively. The temperature distributions in the first and second layers are expressed by $u_{1}\left(x,t\right)$ and $u_{2}\left(x,t\right)$, *t* is the time of the heat transfer process. The temperature distributions in $R_{1}$ and $R_{2}$ are expressed by the following partial differential equations: (1)$$\frac{\partial u_{1}}{\partial t}-\alpha_{1}\frac{\partial^{2}u_{1}}{\partial x^{2}}=0,0<x<l,\;t>0$$
(2)$$\frac{\partial u_{2}}{\partial t}-\alpha_{2}\frac{\partial^{2}u_{2}}{\partial x^{2}}=0,l<x<2l,\;t>0$$

The additional initial value and the boundary conditions are as follows:(3)$$u_{1}\left(x,0\right)=u_{2}\left(x,0\right)=0,\;0<x<2l$$
(4)$$u_{2}\left(2l,t\right)=g\left(t\right),\;t>0$$
(5)$$\frac{\partial u_{2}}{\partial x}\left(2l,t\right)=0,\;t>0$$
(6)$$u_{1}\left(l,t\right)=u_{2}\left(l,t\right),\;t>0$$
(7)$$k_{1}\frac{\partial u_{1}}{\partial x}\left(l,t\right)=k_{2}\frac{\partial u_{2}}{\partial x}\left(l,t\right),\;t>0$$

g(t) in Equation (4) is boundary temperature of the working end. When boundary temperature of the free end is 0 °C, we can use g(t) as junction temperature of the thermocouple.

For similar inverse heat conduction, Shcheglov analyzed the convergence of this problem by hyperbolic perturbation method [20]. For 1D inverse heat conduction in Cartesian systems with two-layer domains, some numerical methods have been proposed [21] but have not been verified in engineering application examples. The purpose of this paper is to obtain the two-layer domain analytic solution for the 1D inverse heat conduction based on the thermocouple during the laser pulse by Fourier transform (FT) [22]. For the serious morbidity of this problem, a regularization strategy was proposed to reconstruct the solution of this problem stably. In order to explain this, we proposed a two-layer domain regularization method based on the analytic solution for the inverse heat conduction. In this method, we used a filtering kernel method to reconstruct stable monolayer solution and provided the selection strategy for optimal regularization parameters. This method is simple but effective.

This paper is composed of the following parts. In the second part, we analyzed the principle and ill-posed inverse heat conduction based on the working principle of thermocouple. In the third part, we used the filtering kernel method to reconstruct the stable solution for the inverse problem under error estimation and provided the convergence estimate. In the fourth part, we reproduced the laser pulse calibration experiment and verified the effectiveness of the proposed method by using experimental samples.

## 2. Principle Analysis of Inverse Problem 

According to the working principle of the thermocouple, we hold that, after the thermocouple’s cold end is compensated, the boundary temperature at the working end g(t) in Equation (3), Section 1, is obtained by the actual output thermoelectric potential difference of the thermocouple, and that it is the output sample of the thermocouple in response to the laser pulse. 

Assume that $g\;\epsilon \;L^{2}\left(0,\infty \right)$, $u_{1}\left(x,t\right)$ and $u_{2}\left(x,t\right)$ are the solutions of the positive problem; then, in the given x ϵ [0,2l], $u_{1}\left(x,\cdot \right)$, $u_{2}\left(x,\cdot \right),$ and their partial differentials $\partial u_{1}\left(x,\cdot \right)/\partial x$, $\partial u_{2}\left(x,\cdot \right)/\partial x$ also belong to space $L^{2}\left(0,\infty \right)$. The solution of the inverse problem is to solve $u_{1}\left(x,t\right)\;\left(0\leq x\leq l\right)$ in $L^{2}\left(0,\infty \right)$ by using these two conditions: the known g(t) and adiabatic boundary x=2l. 

In a dynamic calibration experiment, there is some error in the test results; then, the actual measured value is defined as $g^{\delta }\left(t\right)\;\epsilon \;L^{2}\left(0,\infty \right)$ and meets.
(8)$$\|g^{\delta }-{g\|}_{L^{2}}\leq \delta$$

δ represents measurement error bounds here. Since Cauchy problem is a typical ill-posed problem, any tiny error in experimental sample $g^{\delta }\left(t\right)$ will cause the blow-up of the solution $u_{1}\left(x,t\right)$ within the interval 0≤x≤l. The root of this problem lies in the kernel function in the high frequency part. The ill-posed inverse problem will be analyzed below: 

‖·‖ is expressed as the norm of $L^{2}\left(0,\infty \right)$, we have
(9)$$\|f\|={\left(\int_{0}^{\infty }{\left|f\left(t\right)\right|}^{2}dt\right)}^{\frac{1}{2}}$$Then, we take the FT of $f\left(t\right)\;\epsilon \;L^{2}\left(0,\infty \right)$ as
(10)$$\overset{\wedge }{f}\left(\xi \right)=\frac{1}{2\pi }\int_{0}^{\infty }f\left(t\right)e^{-i\xi t}dt,\;\xi \;\epsilon \;L^{2}\left(0,\infty \right)$$Its inverse transformation is
(11)$$f\left(t\right)=\frac{1}{2\pi }\int_{0}^{\infty }\overset{\wedge }{f\left(\xi \right)}e^{i\xi t}d\xi$$

We first conduct FT on t of the Equations (2)–(6) in the inverse problem in $R_{2}$ and can obtain the following equation: (12)$${\overset{\wedge }{u}}_{2}\left(x,\xi \right)=cosh(\left(2l-x\right)\sqrt{\frac{i\xi }{\alpha_{2}}})\overset{\wedge }{g}\left(\xi \right),\;l\leq x\leq 2l$$
where
(13)$$\sqrt{\frac{i\xi }{\alpha_{2}}}={\left|\xi /\left(2\alpha_{2}\right)\right|}^{\frac{1}{2}}\left(1+isgn(\xi \right))$$
where sgn is a sign function.

Then, the first order partial derivative with respect to x is conducted on Equation (12): (14)$$\frac{\partial {\overset{\wedge }{u}}_{2}\left(x,\xi \right)}{\partial x}=-\sqrt{\frac{i\xi }{\alpha_{2}}}sinh(\sqrt{\frac{i\xi }{\alpha_{2}}}\left(2l-x\right))\overset{\wedge }{g}\left(\xi \right)$$

FT is conducted with respect to ξ in Equations (12) and (14): (15)$$u_{2}\left(x,t\right)=\frac{1}{2\pi }\int_{0}^{\infty }\overset{\wedge }{g}\left(\xi \right)cosh(\left(2l-x\right)\sqrt{\frac{i\xi }{\alpha_{2}}})e^{i\xi t}d\xi$$
(16)$$\frac{\partial {\overset{\wedge }{u}}_{2}\left(x,\xi \right)}{\partial x}=-\frac{1}{2\pi }\int_{0}^{\infty }\overset{\wedge }{g}\left(\xi \right)\sqrt{\frac{i\xi }{\alpha_{2}}}\times sinh(\sqrt{\frac{i\xi }{\alpha_{2}}}\left(2l-x\right))e^{i\xi t}d\xi$$

Next, FT is conducted with respect to t in Equations (1), (3), (6) and (7) in $R_{1}$:(17)$${\overset{\wedge }{u}}_{1}\left(x,\xi \right)=cosh(\sqrt{\frac{i\xi }{\alpha_{1}}}\left(l-x\right))cosh(\sqrt{\frac{i\xi }{\alpha_{2}}}l)\overset{\wedge }{g}\left(\xi \right)+\frac{k_{2}\sqrt{\alpha_{1}}}{k_{1}\sqrt{\alpha_{2}}}sinh(\sqrt{\frac{i\xi }{\alpha_{1}}}\left(l-x\right))\times sinh(\sqrt{\frac{i\xi }{\alpha_{2}}}l)\overset{\wedge }{g}\left(\xi \right),0\leq \;x\leq l$$

Then, the first order partial derivatives with respect to x is conducted on Equation (17):(18)$$\frac{\partial {\overset{\wedge }{u}}_{1}\left(x,\xi \right)}{\partial x}=-\sqrt{\frac{i\xi }{\alpha_{1}}}sinh(\sqrt{\frac{i\xi }{\alpha_{1}}}\left(l-x\right))cosh(\sqrt{\frac{i\xi }{\alpha_{2}}}l)\overset{\wedge }{g}\left(\xi \right)\;-\frac{k_{2}\sqrt{\alpha_{1}}}{k_{1}\sqrt{\alpha_{2}}}\sqrt{\frac{i\xi }{\alpha_{1}}}cosh(\sqrt{\frac{i\xi }{\alpha_{1}}}\left(l-x\right))\times sinh(\sqrt{\frac{i\xi }{\alpha_{2}}}l)\overset{\wedge }{g}\left(\xi \right),\;0\leq x\leq l$$

Then, FT is conducted with respect to ξ in Equations (17) and (18) to obtain $u_{1}\left(x,t\right)$. That is, the numerical solution of the inverse problem is obtained. However, it can be seen from Equations (13) and (18) in $R_{1}$ that the reason for its ill-posedness is the existence of unbounded kernel $cosh(\left(l-x\right)\sqrt{\frac{i\xi }{\alpha_{1}}}),$ which will be amplified infinitely with noise as a factor, thus causing the blow-up of the solution. Therefore, we use the idea of a corrected kernel [23] and provide a bounded kernel to approach unbounded kernel in a new form of filtering kernel. This problem can be solved in this way.

## 3. Filtering Kernel Method

In $R_{1}$, if $\omega =\sqrt{i\xi /\alpha_{1}}$, for the unbounded kernel $cosh(\left(l-x\right)\sqrt{i\xi /\alpha_{1}})$, we provide the filtering form as follows: (19)$${\overset{\wedge }{u}}_{1}{}_{\beta }^{\delta }\left(x,\xi \right)=\frac{cosh(\omega \left(l-x\right))}{1+\beta {\left|\cosh\left(\omega \right)\right|}^{m}}{\overset{\wedge }{g}}^{\delta }\left(\xi \right)$$

If β is a minimal value, $\frac{cosh(\omega \left(l-x\right))}{1+\beta {\left|\cosh\left(\omega \right)\right|}^{m}}\to cosh(\left(l-x\right)\sqrt{\frac{i\xi }{\alpha_{1}}})$, if β>0, then $cosh(\omega \left(l-x\right))/(1+\beta {\left|\cosh\left(\omega \right)\right|}^{m})$ is bounded.

We provide an important lemma:

**Lemma** **1.**
*If a≥0,b≥0, x≥0, ξ ϵ (0,∞), then*
(20)$$\left|cosh\left(a+ibsgn\left(\xi \right)\right)\right|\geq \frac{\sqrt{1-2e^{\frac{-\pi }{2}}}}{2}e^{a}$$
(21)$$\left|\cosh\left(x\left(a+ibsgn\left(\xi \right)\right)\right)\right|\leq e^{xa}$$


The following two theorems indicated that the regular solution is a good approach to the exact solution.

**Theorem** **1.***Assume that*$u_{1}\left(x,t\right)$*is the regular solution of the inverse problem for the exact solution*g(t)*and*$u_{1}{}_{\beta }^{\delta }\left(x,t\right)$*is that for disturbance data*$g^{\delta }\left(t\right)$*; the disturbance data*$g^{\delta }\left(t\right)$*meets*$\|g^{\delta }-{g\|}_{L^{2}\left(R\right)}\leq \delta ,$*and the boundary temperature*$u_{1}\left(l,\cdot \right)$ meets
(22)$$\|u_{1}\left(l,\cdot \right)\|\leq E$$
*Here, E>0 is a given constant; if*
(23)$$\beta ={\left(\frac{\delta }{E}\right)}^{m}$$
*then the following convergence estimation within the length interval 0<x<l can be obtained*
(24)$$\|u_{1}{}_{\beta }^{\delta }\left(x,\cdot \right)-u_{1}\left(x,\cdot \right)\|\leq K\delta^{l-x}E^{x}$$
*where K is a positive constant.*


It is worth noting that in Theorem 1, the convergence estimation is only limited within the interval 0<x<l. However, any useful information on the continuity of the solution cannot be obtained at the boundary x=l. Then, in order to obtain the continuous dependence of the solution at x=l, a stronger transcendental assumption is introduced: (25)$$\|u_{1}{\left(l,\cdot )\right\|}_{P}\leq E$$
where $\left\|\cdot_{P}\right\|$ represents a norm in Sobolev space $H^{P}\left(R\right)$.

**Theorem** **2.**
*Assume that*
$u_{1}\left(x,t\right)$
*is the regular solution of the inverse problem for the exact solution*
g(t),
*and*
$u_{1}{}_{\beta }^{\delta }\left(x,t\right)$
*is that for disturbance data*
$g^{\delta }\left(t\right)$
*; the disturbance data*
$g^{\delta }\left(t\right)$
*meets*
$\|g^{\delta }-{g\|}_{L^{2}\left(R\right)}\leq \delta ,$
*and the boundary temperature $u_{1}\left(l,\cdot \right)$ meets Equation (25); if*
(26)$$\beta ={\left(\frac{\delta }{E}\right)}^{\frac{m}{2}}$$
*then the following convergence estimation can be obtained at*
x=l
(27)$$\|u_{1}{}_{\beta }^{\delta }\left(x,\cdot \right)-u_{1}\left(x,\cdot )\right\|\leq c_{2}\delta^{\frac{1}{2}}E^{\frac{1}{2}}+K_{1}Emax\left\{{\left(\frac{1}{6m}\ln\frac{E}{\delta }\right)}^{-P},{\left(\frac{\delta }{E}\right)}^{\frac{3m-1}{6}}\right\}$$
*where $K_{1}$ is a positive constant.*


Then, Theorem 2 is verified. It is divided into two parts. We can first obtain the following equation with Parseval’s equation and triangle inequality: (28)$$\begin{array}{l} \left\|u_{1\beta }\left(x,\cdot \right)-u_{1}{}_{\beta }^{\delta }\left(x,\cdot \right)\|=\|\overset{\wedge }{u_{1\beta }}\left(x,\cdot \right)-{\overset{\wedge }{u_{1}}}_{\beta }^{\delta }\left(x,\cdot \right)\|=\|\frac{cosh(\omega )}{1+\beta {\left|\cosh\left(\omega \right)\right|}^{m}}\left({\overset{\wedge }{g}}^{\delta }-\overset{\wedge }{g}\right)\right\|\\ \;\leq \delta_{\xi \in R}^{sup}A \end{array}$$

If$$a=b=\sqrt{\frac{i\xi }{\alpha_{1}}},\;\sigma =sgn\left(\xi \right)$$it can be seen that$$A=\frac{cosh(\omega )}{1+\beta {\left|\cosh\left(\omega \right)\right|}^{m}}$$

It can be concluded from Lemma 1 that
$$A\leq \frac{e^{a}}{1+\beta {\left(\frac{1-2e^{-\frac{\pi }{2}}}{4}\right)}^{\frac{m}{2}}e^{ma}}$$

If $f\left(a\right)=\frac{e^{xa}}{1+\beta {\left(\frac{1-2e^{-\frac{\pi }{2}}}{4}\right)}^{\frac{m}{2}}e^{ma}}$ and $c_{1}={\left(\frac{1-2e^{-\frac{\pi }{2}}}{4}\right)}^{\frac{m}{2}}$, it can be obtained by derivation that
(29)$$f^{\prime }\left(a\right)=f\left(a\right)\frac{l-x-\beta c_{1}e^{ma}\left(m-l+x\right)}{1+\beta c_{1}e^{ma}}$$

It is easy to find that(30)$$a_{1}=\frac{1}{m}\ln\frac{l-x}{\beta c_{1}\left(m-l+x\right)}$$meets $f^{\prime }\left(a\right)=0$, besides, when $a>a_{1}$, $f^{\prime }\left(a\right)<0$; when $a<a_{1}$, $f^{\prime }\left(a\right)>0$. Therefore, maximum point of f(a) is $a_{1}$, and
(31)$$f\left(a\right)\leq f\left(a_{1}\right)=c_{2}\beta^{-\frac{l-x}{m}}$$
where $c_{2}=\frac{{\left(l-x\right)}^{\frac{l-x}{m}}}{m}c_{1}{}^{-\frac{l-x}{m}}{\left(m-l+x\right)}^{1-\frac{l-x}{m}}$, thus,
(32)$$A_{1}\leq \frac{e^{\left(l-x\right)a}}{1+\beta c_{1}e^{ma}}\leq c_{2}\beta^{\frac{x-l}{m}}$$

It can be verified in combination with (28) and (32) that
(33)$$\|u_{1\beta }\left(x,\cdot \right)-u_{1}{}_{\beta }^{\delta }\left(x,\cdot \right)\|\leq c_{2}\beta^{\frac{x-l}{m}}$$

Besides,
(34)$$\begin{array}{ll} \left\|u_{1}{}_{\beta }\left(x,\cdot \right)-u_{1}\left(x,\cdot \right)\right\| & =\|{\overset{\wedge }{u_{1}}}_{\beta }\left(x,\cdot \right)-\overset{\wedge }{u_{1}}\left(x,\cdot )\right\|\\ & =\|\left(\frac{cosh(\omega )}{1+\beta {\left|\cosh\left(\omega \right)\right|}^{m}}-cosh(\omega \right))\overset{\wedge }{g}\|\\ & \leq \|\frac{\beta {\left|cosh\left(\omega \right)\right|}^{m}}{1+\beta {\left|cosh\left(\omega \right)\right|}^{m}}{\left(1+{\left|\frac{\xi }{\alpha_{1}}\right|}^{2}\right)}^{-\frac{p}{2}}cosh\left(\omega \right)\overset{\wedge }{g}\|\\ & \leq E_{\xi \in R}^{sup}B \end{array}$$
where $B=\frac{\beta {\left|\cosh\left(\omega \right)\right|}^{m}}{1+\beta {\left|\cosh\left(\omega \right)\right|}^{m}}{\left(1+{\left|\frac{\xi }{\alpha_{1}}\right|}^{2}\right)}^{-\frac{p}{2}}$

It can be concluded from Lemma 1 that
(35)$$B\leq 1+\beta {\left(\frac{1-2e^{-\frac{\pi }{2}}}{4}\right)}^{\frac{m}{2}}e^{ma}{\left(1+{\left|\frac{\xi }{\alpha_{1}}\right|}^{2}\right)}^{-\frac{p}{2}}$$

If $c_{1}={\left(\frac{1-2e^{-\frac{\pi }{2}}}{4}\right)}^{\frac{m}{2}}$, the estimation is conducted under two conditions:

(1) If $a\leq a_{2}=\frac{1}{m}\ln\left(\beta^{-\frac{1}{3m}}\right)$,
(36)$$\leq \frac{\beta e^{ma}}{1+\beta c_{1}e^{ma}}{\left(1+{\left|\frac{\xi }{\alpha_{1}}\right|}^{2}\right)}^{-\frac{p}{2}}\leq \beta e^{ma_{2}}\leq \beta^{1-\frac{1}{3m}}$$

(2) If $a>a_{2}$,
(37)$$B\leq \frac{\beta e^{ma}}{1+\beta c_{1}e^{ma}}{\left(1+{\left|\frac{\xi }{\alpha_{1}}\right|}^{2}\right)}^{-\frac{p}{2}}\;\leq \frac{1}{c_{1}}{\left(1+{\left|\frac{\xi }{\alpha_{1}}\right|}^{2}\right)}^{-\frac{p}{2}}\leq \frac{1}{c_{1}}a_{2}^{-p}=\frac{1}{c_{1}}\left(\frac{1}{m}\ln(\beta^{-\frac{1}{3m}}\right){)}^{-p}$$

Therefore, it can be concluded in combination with (34) and (35) that
(38)$$\begin{array}{ll} \|u_{1}{}_{\beta }^{\delta }\left(x,\cdot \right)-u_{1}\left(x,\cdot )\right\| & \leq \|u_{1}{}_{\beta }^{\delta }\left(x,\cdot \right)-u_{1\beta }\left(x,\cdot \right)\|+\|u_{1\beta }\left(x,\cdot \right)-u_{1}\left(x,\cdot \right)\|\\ & \leq c_{2}\beta^{-\frac{1}{m}}\delta +K_{1}Emax\left\{\left(\left(\frac{1}{m}\ln(\beta^{-\frac{1}{3m}}\right){)}^{-p},\beta^{1-\frac{1}{3m}}\;\right)\right\}\\ & =c_{2}\delta^{\frac{1}{2}}E^{\frac{1}{2}}+K_{1}Emax\left\{\left({\left(\frac{1}{6m}\ln\frac{E}{\delta }\right)}^{-p},{\left(\frac{\delta }{E}\right)}^{\frac{3m-1}{6}}\;\right)\right\} \end{array}$$
where $K_{1}$ is a positive constant. 

We theoretically proved that on the premise of appropriate selection rules for regularization parameters, filtering kernel method is effective. The regular solution for the inverse problem will be obtained by combining experimental samples with the proposed regularization strategy of inverse problem. 

## 4. Calibration System and Experimental Analysis

Although the widely used excitation signal for the dynamic calibration of temperature sensor is temperature step signal, it is difficult to generate ideal step temperature signal, which has brought errors to the dynamic calibration of temperature sensor with step excitation signal, but semiconductor laser can generate very narrow pulse excitation signal. Narrow pulse high-temperature excitation signal is easy to produce and has broader frequency components than step signal; then, it can fully cover the spectrum of thermocouple sensor and excite all of its dynamic characteristics [24]. The RFL-A500D high-power semiconductor laser manufactured by Wuhan Raycus in China is used as the excitation source. Its central wavelength is 915 nm, and its output power is 515 W and is adjustable. The laser pulse calibration system is constructed in this way. See Figure 3 for the composition of the calibration system. 

Figure 4 shows our experimental environment; the thermocouple calibrated is a K-type filament thermocouple with a junction diameter of 1 mm; its measuring range is 0–1200 °C. The conductors are made of nickel and chromium, (nickel: thermal conductivity is 90.0 W/m·k, thermal capacity is 0.46 kJ/kg·°C and density is 8.9 g/cm^3^; chromium: thermal conductivity is 93.7 W/m·k, thermal capacity is 0.45 kJ/kg·°C and density is 7.19 g/cm^3^). The filament thermocouple is fixed on the bracket. Uniform laser beams [25,26,27] are formed after laser pulses converge in the laser flux uniformity system to heat the surface of thermocouple junction. The fiber optic probe of high speed infrared thermometer (the fast response time is 6 (*μs*)) is aligned to the surface of thermocouple junction. The measured temperature signal is used as a response signal g(t) that acts as a laser pulse on the junction surface. The thermocouple potential difference signal is amplified by an amplifier; and the thermocouple laser pulse response signal g(t) and infrared thermometer output signal are synchronously collected by the data acquisition and processing system. Besides, the digital cold end compensation processing is conducted on the free end of the thermocouple, with the output signal of the infrared thermometer as the exact solution $u_{1}\left(l,t\right).$

The output power of the laser is adjusted by percentage; thus, we set the output power of the laser to 90%. The theoretical maximum temperature at 90% power should not be higher than 1000 °C. The pulse width is set to 5 ms. The sampling frequency of data acquisition and processing system is 1 MHz. Both butt-welded and ball-welded thermocouples are excited at an ambient temperature of 21 °C. The pulse response data and junction surface temperature data of the thermocouples at 90% powers are obtained. See Figure 5 (the following operation results and charts are obtained in MATLAB).

It can be seen that peak value of infrared temperature is 694.8 °C and the rise time is 12.5 ms. Different response times of the two junction structures are obtained and both are slower than infrared signal, and there is large deviation in the peak value of temperature.

The following formula is used to generate error data.
(39)$$g^{\delta }\left(t\right)=g\left(t\right)+\sigma \cdot randn\left(size\left(g\left(t\right)\right)\right)$$
where t is sample time, determined by sample size and sampling rate, $t_{i}=\left(i-1\right)\Delta t$, Δt=1/(n−1), i=1,2,⋯,n. According to Equation (8) in Section 2, the value of δ at the junction is:(40)$$\delta :=\|g^{\delta }\left(t\right)-g{\left(t)\right\|}_{L^{2}}=\sqrt{\frac{1}{n}\overset{n}{\underset{i=0}{\sum }}{\left(g_{i}^{\delta }-g_{i}\right)}^{2}}$$

On the basis of Equations (26), (27), (39) and (40) in Section 3, $\beta ={\left(\frac{\delta }{E}\right)}^{\frac{m}{2}}$ is used to take different m values under different σ conditions to obtain a series of regular solutions $u_{1}{}_{\beta }^{\delta }$; the value of E is the maximum range of thermocouple; σ is respectively 0.01, 0.001, 0.0001, and m is integer from 0 to 20. We first observed the changes in δ as shown in Figure 6a,b.

It can be seen from Figure 6 that the two structures have the same tendency, and the smaller the σ value, the more obvious the attenuation. However, when σ values are different, the smaller the σ value, the faster the convergence. When m≥8, σ=0.0001, the error attenuation tends to be stable first, which indicates that the regularization parameters selected tend to be optimal. 

When σ=0.0001, m is 2, 5 and 8, respectively; the regular solution and exact solution under 90% power output are as shown in Figure 7a–d.

The calculation results under different conditions are acquired and listed in Table 1 and Table 2.

It can be seen from Figure 7 and Table 1 and Table 2 that when the value of m is small, although the regular solutions of two junction structures have some distortion, both of them can still approach the exact solution. Furthermore, when the regularization parameters reach the optimal value, the regular solution and the exact solution are very close, we calculated the relative error of butt-welded signal between the regular solution and the exact solution in the pulse region when σ=0.0001 and m=8. See Figure 8. 

The maximum relative error is 7.64%; the relative error of peak value is even less; thus, the proposed regularization method is effective for recovering the junction surface temperature of filament thermocouple.

## 5. Conclusions

Since the thermocouple’s dynamic characteristics are affected by its junction structure and cannot faithfully recover the actual temperature of the measured medium, we constructed a laser pulse calibration system to quantitatively analyze its dynamic characteristics and describe the problem as the solution for 1D inverse heat conduction from the point of view of inverse filtering. For thermocouple of a two-layer conductor structure, this problem cannot be solved by most regularization methods in monolayer domain. We can use inverse heat conduction in stable monolayer domain to extend and solve the two-layer domain problem. We find the kernel function, which causes the instability of the numerical solution in monolayer domain, and use the stable kernel function to filter it to provide the error estimation from the two-layer domain filtering kernel method. With the experimental data from laser pulse calibration, we obtain the optimal regularization parameter, verify that the proposed regularization method can effectively recover the surface temperature of the thermocouple, and make up the deficiency for the dynamic characteristics of the thermocouple methodologically.

## Figures and Tables

**Figure 1 micromachines-11-00044-f001:**
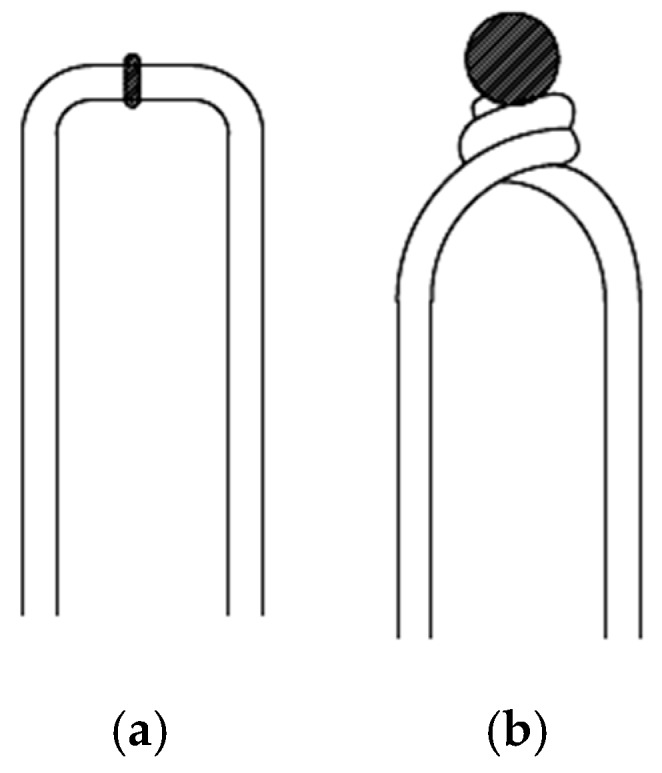
Two types of filamentous thermocouple: (**a**) butt-welded and (**b**) ball-welded.

**Figure 2 micromachines-11-00044-f002:**
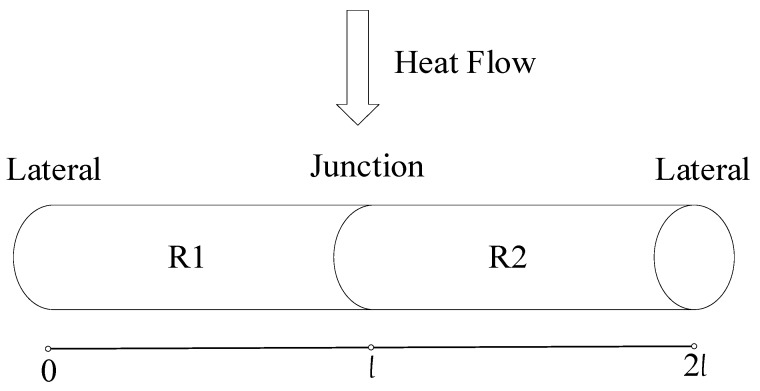
Filament thermocouple.

**Figure 3 micromachines-11-00044-f003:**
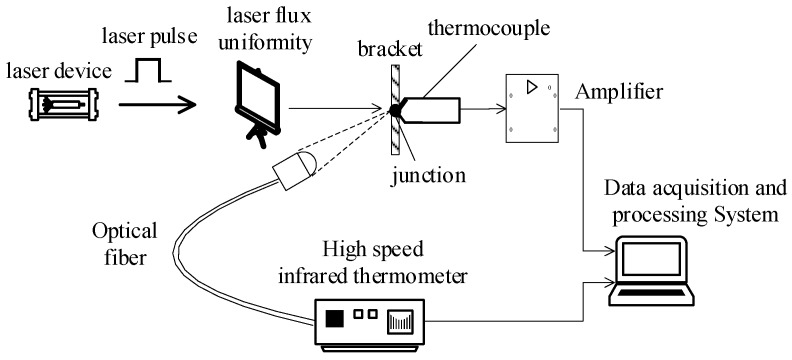
Transient temperature calibration system.

**Figure 4 micromachines-11-00044-f004:**
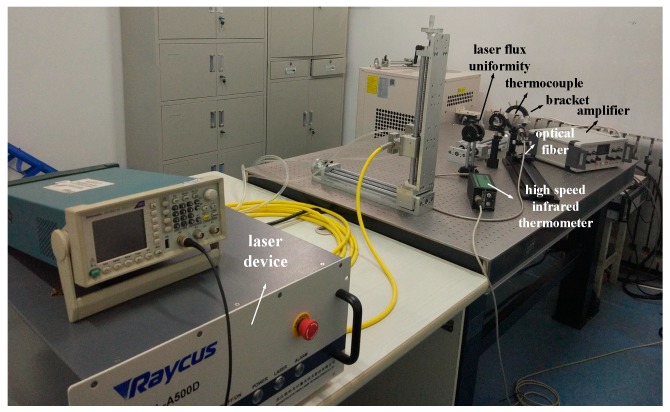
Experimental environment.

**Figure 5 micromachines-11-00044-f005:**
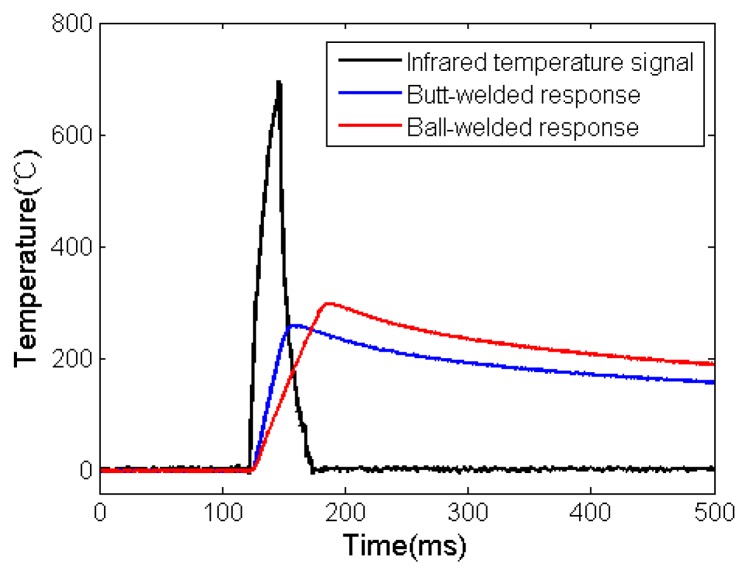
Output signal with two junction structures at 90% power.

**Figure 6 micromachines-11-00044-f006:**
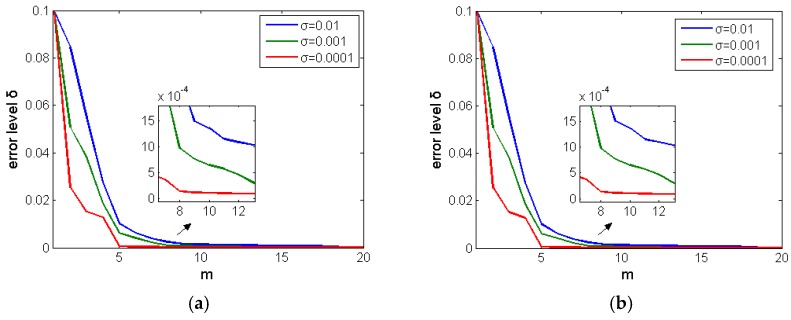
Attenuation of *δ* with the gradual increase of m when σ is 0.01, 0.001 and 0.0001: (**a**) butt-welded signal and (**b**) ball-welded signal.

**Figure 7 micromachines-11-00044-f007:**
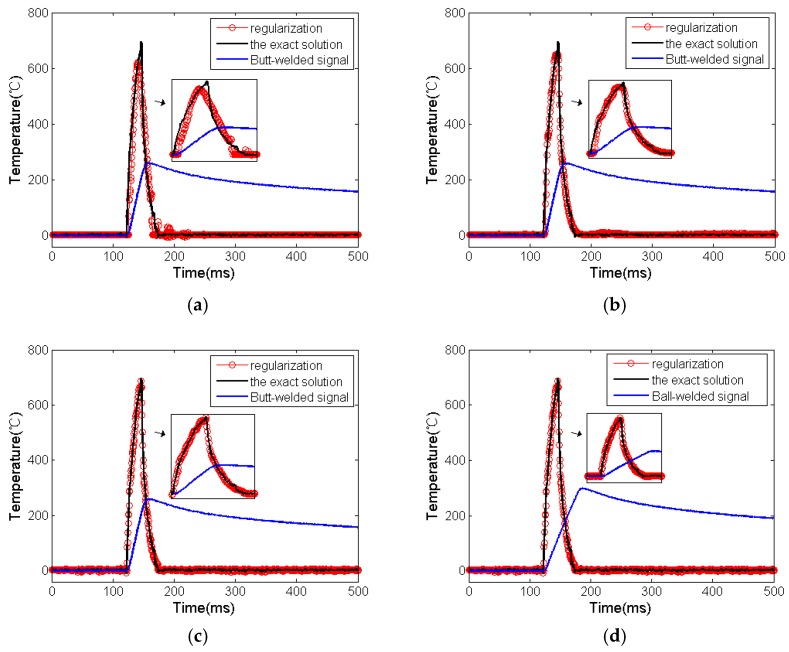
Calculation results of two junction structures: (**a**) butt-welded signal, exact solution and regular solution when σ=0.0001, m=2; (**b**) butt-welded signal, exact solution and regular solution when σ=0.0001, m=5; (**c**) butt-welded signal, exact solution and regular solution when σ=0.0001, m=8; (**d**) ball-welded signal, exact solution and regular solution when σ=0.0001, m=8.

**Figure 8 micromachines-11-00044-f008:**
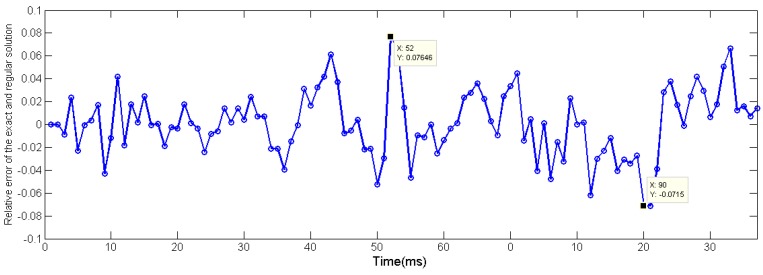
Relative errors between regular solution and exact solution.

**Table 1 micromachines-11-00044-t001:** *σ* = 0.0001 and *m* = 2, 5, 8; the peak value and rise time of butt-welded thermocouple after regularization.

Title 1	Original	*M* = 2	*M* = 5	*M* = 8
Peak value (°C)	262.1	621.6	674.5	692.5
Rise time (ms)	25.9	13.7	12.7	12.5

**Table 2 micromachines-11-00044-t002:** *σ* = 0.0001 and m = 8; the peak value and rise time of butt-welded thermocouple after regularization.

Title 1	Original	*M* = 8
Peak value (°C)	304.4	691.2
Rise time (ms)	43.2	12.5

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
