# Peer review of "Dynamic Error Correction of Filament Thermocouples with Different Structures of Junction based on Inverse Filtering Method"

_micromachines, 2019, doi:10.3390/mi11010044_

Round 1

Reviewer 1 Report

The manuscript was concerning a point that has been going for long with accuracy of commercial grade thermocouples, and k-type TCs were tested and validated with its responsive and accuracy, based on the junction methods.

However, the given manuscript has not successfully presented the fulfilled work and impact of them, so the introduction and abstract may need to be revised. 

The reviewer criticises,

line 39: 'analysis' is typo? line 87~: some key factors were not identified well, including, t, the location of the heat source (so the reviewer has failed to clarify the conduct heat flow direction), the given × are identical in both of R1 and R2? line 119: g(t) appeared all of sudden? rest of the given equations are subject to the above comment to be accepted. line 254, Fig 3 font size is too small, and not showing the conductor (line 259) or the braket (line 260). line 260~264: needs to address the device/equipment details line 266: g(t) appeared again, and is this term associated with the earlier g(t) on line 119? Further description of identifying the co-relation to the other equations is required. line 270-271: needs to be slimed down line 316: the given figure, ie 8%, is not evidently supported, isn't it? The manuscript has not addressed the aspect of heat capacity and thermal inertia of the k-type TCs. The reviewer is not convinced to agree the 'quantitatively analyse its dynamic characteristics', line 321, without studying such. the reference list: only 4 out of 21 references are published on 2016 or afterwards whilst the manuscript should have covered more of recent key references. The ref list should include recent key publications of single and multi-channel thermocouple study.

Reviewer 2 Report

The regularization mechanism should be better explained and presented on the chart. This is the essence of the approach proposed by the authors.

The authors mention that: “The regularization parameter selection strategy proposed in the Theorem 2 in Section 3 is used..”, but in this section is described that: “appropriate selection rules for regularization parameters…is effective” (line 240).

In the reviewer's opinion, the “appropriate selection” should be clearly specified.

There are a few errors in references. The author of the third position is: first name Magdalena, last name Jaremkiewicz so it should be: Jaremkiewicz M.

There is probably an error in position [19]. The reviewer could not find this publication: Li Y., Zhang Z., Hao X., et al. A Dynamic calibration method of thermocouple sensor based on laser. Sensors, 2018, 18, 1-14.

Round 2

Reviewer 1 Report

The revised version has been largely met the standard of this journal, so should be published. However, it should be also commented that the reviewer is not entirely satisfied with the provided reference list; quite a few of recent key publications were not included in. Based on the reviewer's previous comment, the authors have had put a single journal report only to the initial reference list. The authors must revisit this list and faithfully address the criticism. 
